# Systematic Review and Critical Evaluation of Quality of Clinical Practice Guidelines on Nutrition in Pregnancy

**DOI:** 10.3390/healthcare10122490

**Published:** 2022-12-09

**Authors:** Marika De Vito, Sara Alameddine, Giulia Capannolo, Ilenia Mappa, Paola Gualtieri, Laura Di Renzo, Antonino De Lorenzo, Francesco D’ Antonio, Giuseppe Rizzo

**Affiliations:** 1Department of Obstetrics and Gynecology Fondazione Policlinico Tor Vergata, Università di Roma Tor Vergata, 00185 Roma, Italy; 2Department of Obstetrics and Gynecology, Università di Chieti, 66100 Chieti, Italy; 3Department of Biomedicine and Prevention, Università di Roma Tor Vergata Section of Clinical Nutrition and Nutrigenomic, 00185 Roma, Italy

**Keywords:** nutrition in pregnancy, weight gain in pregnancy, clinical practice guidelines

## Abstract

Objective: To report the quality and clinical heterogeneity of the published clinical practice guidelines (CPGs) on nutrition in pregnancy. Methods: MEDLINE, Embase, Scopus, and ISI Web of Science databases were searched. The following aspects related to nutrition in pregnancy were addressed: specific requirements during pregnancy, description of a balanced diet, weight gain, prevention of food-borne, nutrition in peculiar sub-groups of women, and maternal or perinatal outcomes. The assessment of the risk of bias and quality assessment of the included CPGs were performed using “The Appraisal of Guidelines for REsearch and Evaluation (AGREE II)” tool divided in six quality domains: scope and purpose, stakeholder involvement, rigor of development, clarity of presentation, applicability, editorial independence. Mean ± standard deviation (SD) was used to summarize the scores across all the guidelines per domain. The quality of each guideline was computed using the scoring system proposed by Amer et al. A cut-off of >60% was sued to define a CGP as recommended. Results: Eighteen CPGs were included. There was a substantial heterogeneity in the recommended dose for vitamins, folic acid, and micronutrient intake during pregnancy among the different published CPGs. 27.8% (5/18) of the CPGs recommended a daily intake of folic acid of 200 mcg, 38.8% (7/18) 400 mcg, 16.7% (3/18) 600 mcg while the remaining CPGs suggested dose between 400 and 600–800 mc per day. Adequate maternal hydration was advocated in the large majority of included CPGs, but a specific amount of water intake was not reported in 83.3% (15/18) cases. There was also significant heterogeneity in various other aspects of nutrition recommendation among the different CPGs, including gestational weight gain (55.5%), prevention of food-borne diseases in pregnancy (72.2%), nutrition in particular groups of pregnant women (83.3%), maternal and perinatal outcomes (72.2%). The AGREE II standardized domain scores for the first overall assessment (OA1) had a mean of 65% but only half scored more than 60%. Conclusion: The published CPGs on nutrition in pregnancy show an overall good methodology, but also a substantial heterogeneity as regard as different major aspects on nutrition in pregnancy.

## 1. Introduction

Pregnancy represents a crucial period in women’s life, which influences the future health of both mother and children [1].

A balanced diet and a proper weight gain during pregnancy are associated with better maternal and perinatal outcomes. Conversely, both over and under-nutrition carry a higher risk of adverse perinatal outcomes, including impaired or excessive fetal growth, gestational diabetes mellitus (GDM), preterm birth (PTB), and pre-eclampsia (PE). More importantly, lack of adequate nutrition can cause long-term consequences later in life [2,3,4,5].

Therefore, weight gain and nutrition represent a key public health issue, and several clinical practice guidelines (CPGs) on nutrition and adequate weight gain during pregnancy have been released by the national and international body societies in the last few years [6,7,8,9,10,11,12].

CPGs are documents including recommendations aimed to improve patient care. Generation of CPGs require a rigorous methodology allowing to provide clinicians with the most up-to-date and objective clinical evidence. The Appraisal of Guidelines for REsearch and Evaluation tool (AGREE II) is the most commonly utilized tool to test the quality of CPGs, and it is considered the “gold standard” for CPG quality assessment. We used this methodology instead of AGREE- REX tool on the basis of previous studies on nutrition comparing both methods [13,14]. The published literature on nutrition in pregnancy is heterogeneous on many aspects of nutrition and weight gain in pregnancy, which likely reflects geographical differences and nutritional plans among the different countries. The question that arise is to evaluate the homogeneity of nutritional guidelines for pregnant women both in term of contents and methodology.

The primary objective of this systematic review was to objectively evaluate the methodological quality of the published CPGs on nutrition in pregnancy using the AGREE II methodology during pregnancy. The secondary objective was to evaluate the clinical heterogeneity among the different CPGs regarding different aspects of the nutrition and weight gain in pregnancy.

## 2. Materials and Methods

### 2.1. Protocol, Information Sources, Literature Search, and Data Extraction

This review was designed following a recommended protocol for Systematic Review of Clinical Guidelines and following Prisma guidelines (Appendix A). The literature search was performed in the MEDLINE (PubMed), Scopus, and ISI Web of Science databases to identify all CPGs on nutrition and weight gain in pregnancy published before and between January 2010 and June 2022. Combinations of the following keywords and MESH search terms were used: “nutrition”, “weight gain” “pregnancy”, and “clinical practice guidelines”. Both “AND” and “OR” were used as search Boolean functions.

Inclusion criteria were all English-written CPGs on nutrition and weight gain in pregnancy as described in PICAR (Population and clinical areas, Interventions, Comparator, Attributes of eligible CPGs and Recommendation characteristics) statement (Table 1, Figure 1) [15].

Four reviewers (MDV, IM, GC, and SA) independently evaluated titles and abstracts. Disagreements were resolved by discussion among authors, and if necessary, with the participation of two senior authors (GR, FDA). When more than one version of the same CPG was available, only the last updated version was considered eligible for the inclusion. The main data extracted for the present review included publication ID (first author, a research consortium, or a professional society), year of publication, country, title, society, scope of the CPG, publication date, number of revisions, and type of methodology adopted. The outcomes were extracted and reported in an online Google sheet for sharing among all authors.

### 2.2. Outcomes Measures

The following aspects related to the f nutrition in pregnancy were considered:Specific nutritional requirements;Description of balanced diet before and through pregnancy;Optimal weight gain;Prevention of food-borne diseases;Nutrition in peculiar sub-groups of women;Maternal outcomes, including anemia, caesarean section, and mortality;Perinatal outcomes, including SGA, perinatal mortality, preterm birth, and congenital anomalies.

### 2.3. Quality Appraisal of Guidelines and Risk of Bias

“The Appraisal of Guidelines for Research and Evaluation (AGREE II)” tool was used to assess the risk of bias and quality assessment of the included CPGs [16,17]. The AGREE II tool includes 23 items divided into six quality domains:Scope and purpose;Stakeholder involvement;Rigor of development;Clarity of presentation;Applicability;Editorial independence.

Each of the items evaluates different aspects of the quality of the practice guideline. Each item was rated on a seven-point scale ranging between 1 (strongly disagree) to 7 (strongly agree). A final overall assessment was performed including the evaluation of the overall quality of the CPG (OA1) and if the CPG would be recommended to be applied in practice (OA2). To start the appraisal process, it is recommended that at least two investigators analyze each clinical guideline to increase the reliability of the evaluation. The standardized domain score would be 0% if each investigator scored 1 for all the variables included in this domain (https://www.agreetrust.org/resource-centre/agree-ii) (accessed on 12 October 2022). The methodology of reaching consensus was used to score the items. After considering all the 23 items and obtaining the comprehensive judgment of the reviewers, CPGs evaluation was grouped into three categories according to the AGREE II score, (recommended, recommended after revision, and not recommended).

### 2.4. Statistical Analysis

Statistical analysis was carried out as descriptive statistics. We calculated frequencies and raw proportions to summarize the main recommendations among nutrition and weight gain in pregnancy. Moreover, we calculated the quality of CPGs using AGREE II domain scores, as AGREE II recommended (https://www.agreetrust.org/resource-centre/agree-ii) (accessed on 12 October 2022). Mean ± standard deviation (SD) was used to summarize the scores across all the guidelines per domain. The AGREE II tool does not provide any advice on how to define a quality score. However, to define a CPG as of good quality we adopted the cut-off score of 20 according to Amer et al.; if the overall guideline score was >60%, the CPG was recommended; if the overall guideline score was between 40% and 60%, the CPG was recommended after modification; and if the guideline score was <40%, it was not recommended. The analysis was performed using Excel 16.57 (© 2021 Microsoft Corporation. Hongkong, China, All rights reserved.) software and SPSS (version 26.0 IBM Corp. Armonk, NY, USA) software.

## 3. Results

### 3.1. Study Selection and General Characteristics

A total of 43 articles were identified, and 18 CPGs were included in the analysis [18,19,20,21,22,23,24,25,26,27,28,29,30,31,32,33,34,35] (Table 2). The documents produced by the American Dietetics Association [36] and American College of Obstetrician and Gynecologist [37] were excluded since they are structured as position paper or frequently asked questions for women and not as guidelines.

The CPGs analyzed in this SR were characterized by GRADE and GRADE-CERQual approaches and DECIDE framework, Review of literature, expert opinion, and expert panel consensus. 4/18 (WHO, The New York Academy of Science, Polish Society of Gynecologists and Obste-tricians and Australian Government, Department of Health and Ageing) were published in 2020, 3/18 (The Early Nutrition Project partners, Alberta Health Services and The Public Health Division of the Pacific Community) in 2019; 2/18 (Fondazione Confalonieri Ragonese and The Republic of the Union of Myanmar, Ministry of Health and Sports) in 2018; 1/18 (RIGA-WHO) in 2017; 1/18 (Italian Consensus document) in 2016; 1/18 (FIGO) in 2015; 1/18 (Institute of Obstetricians and Gynecologists) was first published in 2013 and revised in 2019; 1/18 (Philippine Obstetrical and Gynecological Society) was first published in 2013 and revised in 2018; 1/18 (Family Health Bureau, Ministry of Health) was published in 2011; 1/18 (Minister of Health, Canada) was published in 2009; 1/18 (NICE) was first published in 2008 and revised in 2022; 1/18 (Ministry of Health of New Zealand) was first published in 2006 and revised in 2008.

### 3.2. Synthesis of the Results

There was a substantial heterogeneity in the recommended dose for vitamins intake during pregnancy among the different published CPGs with no specific recommendation in about 40–50% of them. Likewise, there was also substantial heterogeneity in the recommended dose of folic acid and micronutrient intake. 27.8% (5/18) of the CPGs included in the present systematic review recommended a daily intake of folic acid of 200 mcg, 38.8% (7/18) 400 mcg, 16.7% (3/18) 600 mcg, while the remaining CPGs suggested a dose between 400 and 600–800 mc per day. Although adequate maternal hydration was generally recommended by all the included CPG, a specific amount of water intake was not reported in 83.3% (15/18) of the cases. There were also substantial heterogeneity as regard as several clinical points related to nutrition in pregnancy, including gestational weight gain (55.5%), prevention of food-borne diseases in pregnancy (72.2%), nutrition in particular groups of pregnant women (83.3%), maternal outcomes (72.2%), and fetal/neonatal outcomes, in particular SGA (61.1%), low birthweight (72.2%), preterm birth (72.2%), perinatal mortality (88.9%), neonatal mortality (83.3%), stillbirth (83.3%), and congenital anomalies (83.3%). Appendix A describes different topics analyzed among other CPGs studied in this systematic review.

Table 3 describes a synthesis of all the AGREE II domains and shows the average standardized score for each domain.

The AGREE II standardized domain scores for the first overall assessment (OA1) had a mean of 65%. Half of CPGs analyzed (9/18) were rated more than 60% and indicate a consensus agreement between the reviewers on recommending the use of these CPGs.

## 4. Discussion

### 4.1. Main Findings

The findings from this systematic review of CPGs showed that the majority of CPGs on nutrition and weight gain in pregnancy are of generally good quality. However, there was substantial heterogeneity as regard several key points, including vitamins, folic acid, and micronutrients dose, hydration, and the reported risk of adverse maternal and fetal outcomes related to inappropriate diet [38,39,40,41].

### 4.2. Strengths and Limitations; Comparison with Other Systematic Reviews

This is, to the best of our knowledge, the first systematic review exploring the methodological quality and clinical heterogeneity of CPGs on nutrition and weight gain in pregnancy using the AGREE-II tool. Other strengths included the thorough literature search and assessment of the different aspects of nutrition.

The main weakness of the present review relies on the inclusion of CPGs written only in the English language and the lack of assessment of CPGs reporting pre-pregnancy nutrition.

### 4.3. Clinical and Research Implications

Appropriate maternal nutrition in pregnancy is crucial. Both over and undernutrition are associated with sub-optimal pregnancy outcome and metabolic derangement in both the mother and off-springs. More recently, a better maternal diet quality during pregnancy has been demonstrated to have a small positive association with child neurodevelopment, with more reliable results seen for cognitive development.

In view of its importance, it is not surprising that many national and international body societies have released CPGs on the type of optimal nutrition and weight gain in pregnancy.

A healthy and varied diet is the preferred mean to achieving the optimal nutritional requirements in pregnancy. However, some nutritional needs in pregnancy are difficult to meet with diet alone, thus requiring micronutrient supplement use and food fortification programs. About 20% to 30% of pregnant women worldwide suffer from vitamin deficiency. Optimal folic acid intake of folic acid is crucial both before and during pregnancy, in view of its association with a reduced risk of neural tube defects. Folic acid is essential for several other biological functions, including cell replication, division, and survival. It is inactive in the human body and must be converted by the liver to the active molecule 5-methyltetrahydrofolate (5-MTHF) methylenetetrahydrofolate reductase (MTHFR), methionine synthase (MTR), and methionine synthase reductase (MTRR) which play pivotal roles in the folic acid cycle, using as cofactors vitamin B6 and vitamin B12.

Common polymorphisms of MTHFR (C677T and A1298C), MTRR (A66G) and MTR (A2756G) enzymes may influence the serum folate level and contribute to folate deficiency [38,39]. Moreover, impaired DNA methylation of MTHFR, as hypermethylation, has been related to reduced gene expression, contributing to several human disorders [42]. Some pregnancy diseases, such as megaloblastic anemia due to vitamin B12 deficiency, could be avoided by supplementing 5-MTHF instead of folic acid [43,44].

In the present systematic review, we also found substantial heterogeneity in the recommended dose of iron and other micronutrient supplementation in pregnancy, with many CPGs not reporting the actual dose of such nutrients.

Optimal iron supplementation in pregnancy is also fundamental to reduce the risk of fetal anemia, especially in women close to delivery. Anemia due to iron deficiency is the most common cause of anemia in pregnancy, particularly in developed countries [43], and carries a high susceptibility to infection, increased risk of peripartum transfusion, pre-eclampsia, placental abruption, etc., [45].

Women should be extensively counselled about the need for a varied and balanced diet during pregnancy in order to maximize short and long-term maternal and perinatal outcomes [46,47,48]. Furthermore, because pregnancy is not commonly planned, it is also necessary to raise awareness of the importance of a balanced diet and correct lifestyle among women of childbearing age [49].

## 5. Conclusions

The published CPGs on nutrition and weight gain in pregnancy are characterized by an overall good methodology but a substantial clinical heterogeneity. The findings from this systematic review support the need for CPGs shared by the different national and international body societies aimed at providing homogenous recommendations regarding nutrition and weight gain in pregnancy, also considering geographical differences and diet plans among the different populations.

## Figures and Tables

**Figure 1 healthcare-10-02490-f001:**
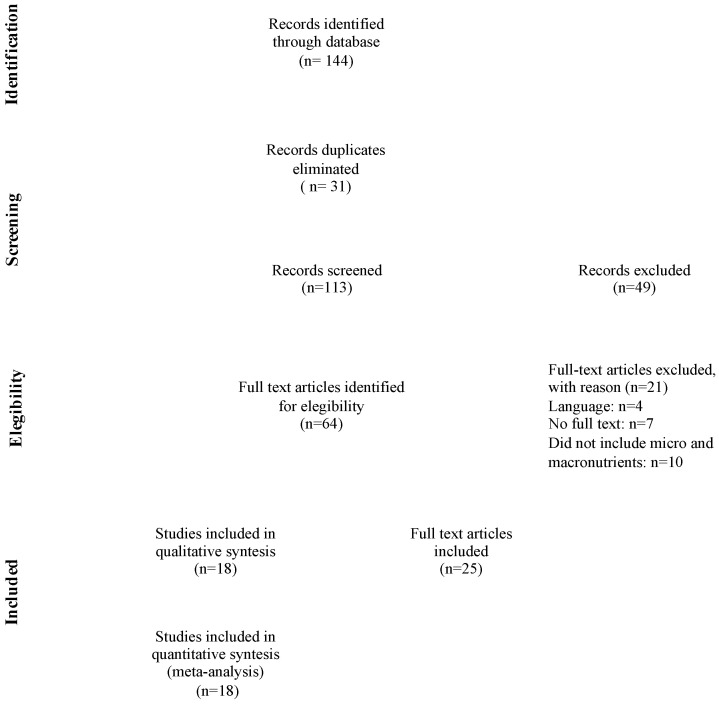
PRISMA flow diagram.

**Table 1 healthcare-10-02490-t001:** PICAR statement for inclusion of CPGs.

Criterion	Description
(P) Population	Pregnant women
(I) Interventions	Any nutrition and dietary intervention for a healthy pregnancy (description of micro and macro nutrients, GWG, etc.)
(C) Comparators	Any comparator or comparison. No key CPG content is of interest
(A) Attributes of eligible CPGs	1. Local, national, and international CPGs, including consensus panel 2. Only full-text articles available 3. Issued from professional and/or governmental organizations 4. Published since 2000 until 30 May 2022 5. Articles and CPGs in English language 6. Latest version 7. No geographic limitation 8. Reporting nutrition and GWG-related recommendations 9. Intended for health professionals 10. No restrictions on quality, as assessed by the AGREE II instrument
(R) Recommendation characteristics and other considerations	Not applicable

**Table 2 healthcare-10-02490-t002:** General characteristics of CPGs included.

Guideline	Country	Year	Last Revision	Scope	Methodology
WHO	International	2020	2020	International	GRADE and GRADE-CERQual approaches, DECIDE framework
RIGA-WHO	Latvia	2017	2017	International and National	Review of literature, expert opinion
FIGO	International	2015	2015	International	Review of literature, expert panel consensus
The Early Nutrition Project partners	International	2019	2019	National	Review of literature, expert panel consensus
NICE	England	2008	2022	National	Review of literature, expert opinion, and formal consensus
The New York Academy of Science	USA	2020	2020	National	Review of literature, expert panel consensus
Fondazione Confalonieri Ragonese	Italy	2018	2018	National	Review of literature, expert opinion
Italian consensus Document	Italy	2016	2016	National	Review of literature, expert panel consensus
Institute of Obstetricians and Gynecologists	Ireland	2013	2019	National	Review of literature, expert opinion
Polish Society of Gynecologists and Obstetricians	Poland	2020	2020	National	Review of literature, expert opinion
Australian government, Department of Health and Ageing	Australia	2020	2020	National	Review of literature, expert opinion
Ministry of Health of New Zealand	New Zeeland	2006	2008	National	Review of literature, expert opinion
Minister of Health, Canada	Canada	2009	2009	National	Review of literature, expert opinion
Alberta Health Services	Canada	2019	2019	Local	Review of literature, expert opinion
Philippine Obstetrical and Gynecological Society	Philippines	2013	2018	National	Review of literature, expert opinion
The Public Health Division of the Pacific Community	Pacific Community- Noumea, New Caledonia	2019	2019	National	Review of literature, expert opinion
The Republic of the Union of Myanmar, Ministry of Health and Sports	Myanmar	2018	2018	National	Review of literature, expert opinion
Family Health Bureau, Ministry of Health	Sri Lanka	2011	2011	National	Review of literature, expert opinion

**Table 3 healthcare-10-02490-t003:** Summary of all the AGREE II domains.

RD ID Agree II	Domain 1 (Items 1–3)	Domain 2 (Items 4–6)	Domain 3 (Items 7–14)	Domain 4 (Items 15–17)	Domain 5 (Items 18–21)	Domain 6 (Items 22–23)	OA 1	OA2
Canada	100%	76%	71%	100%	82%	93%	100%	Y (*n* = 2) YWM (*n* = 0) N (*n* = 0)
The Early Nutrition Project partners	81%	57%	39%	33%	54%	79%	29%	Y (*n* = 0) YWM (*n* = 0) N (*n* = 2)
The New York Academy of Sciences	95%	76%	61%	48%	75%	79%	86%	Y (*n* = 1) YWM (*n* = 1) N (*n* = 0)
FIGO	100%	90%	88%	95%	71%	100%	100%	Y (*n* = 2) YWM (*n* = 0) N (*n* = 0)
Myanmar	90%	57%	39%	29%	54%	57%	29%	Y (*n* = 0) YWM (*n* = 0) N (*n* = 2)
NICE	100%	95%	86%	48%	43%	100%	57%	Y (*n* = 0) YWM (*n* = 2) N (*n* = 0)
Polish	100%	95%	84%	100%	75%	100%	57%	Y (*n* = 0) YWM (*n* = 2) N (*n* = 0)
Ireland	81%	67%	68%	86%	82%	86%	86%	Y (*n* = 1) YWM (*n* = 1) N (*n* = 0)
RIGA	62%	48%	45%	62%	47%	86%	43%	Y (*n* = 0) YWM (*n* = 1) N (*n* = 1)
Sri Lanka	95%	76%	77%	62%	36%	100%	71%	Y (*n* = 0) YWM (*n* = 2) N (*n* = 0)
WHO	95%	76%	96%	95%	75%	93%	100%	Y (*n* = 2) YWM (*n* = 0) N (*n* = 0)
Alberta	90%	81%	73%	90%	68%	50%	86%	Y (*n* = 2) YWM (*n* = 0) N (*n* = 0)
Confalonieri Ragonese	76%	81%	79%	86%	81%	36%	86%	Y (*n* = 2) YWM (*n* = 0) N (*n* = 0)
Italian Consensus	95%	76%	66%	81%	54%	79%	43%	Y (*n* = 0) YWM (*n* = 1) N (*n* = 1)
New Caledonia	81%	67%	32%	57%	50%	29%	29%	Y (*n* = 0) YWM (*n* = 0) N (*n* = 2)
Philippines	57%	48%	61%	86%	64%	79%	43%	Y (*n* = 0) YWM (*n* = 1) N (*n* = 1)
New Zealand	81%	86%	61%	86%	64%	64%	71%	Y (*n* = 0) YWM (*n* = 2) N (*n* = 0)
Australia	100%	76%	79%	48%	39%	100%	57%	Y (*n* = 0) YWM (*n* = 2) N (*n* = 0)
Average score for each domain (n%)	88%	74%	67%	72%	62%	78%	65%	
SD for each domain (±%)	13%	33%	18%	23%	15%	22%	25%	

Green color: recommended; yellow color: revised recommended; red color: not recommended.

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
