# Peer review of "Systematic Review and Critical Evaluation of Quality of Clinical Practice Guidelines on Nutrition in Pregnancy"

_healthcare, 2022, doi:10.3390/healthcare10122490_

Round 1

Reviewer 1 Report

Brief Summary

The manuscript is a systematic review that aimed to report the quality and clinical heterogeneity of clinical practice guidelines for nutrition during pregnancy. The main contribution of the manuscript is to review nutrition clinical recommendation and strength is to explore recommendation in a systematic way.

General concept comments

Article review

The manuscript needs a detailed review of English writing and readability. Most phrases seem incomplete with typos throughout the text. Revision of the language is needed for comprehension of the manuscript. Additionally, for the systematic review approach the authors did not present search terms Boolean function. I also missed clinical guidelines well-known, such as American Dietetics Association, ACOG, and WHO.

Author Response

The manuscript is a systematic review that aimed to report the quality and clinical heterogeneity of clinical practice guidelines for nutrition during pregnancy. The main contribution of the manuscript is to review nutrition clinical recommendation and strength is to explore recommendation in a systematic way.

Thanks for the positive comment

The manuscript needs a detailed review of English writing and readability. Most phrases seem incomplete with typos throughout the text. Revision of the language is needed for comprehension of the manuscript

We apologize. The new version has been revised by a native english speaker

Additionally, for the systematic review approach the authors did not present search terms Boolean function

Thanks for the suggestion

MESH criteria were already and we specified search boolean function used (lines 

I also missed clinical guidelines well-known, such as American Dietetics Association, ACOG, and WHO.

We did not included thee American Dietics Association document since it is  a a position paper and not a guideline. 

Similarly ACOG produced FAQ for women on nutrition  and cannot be considered a guideline

These criteria are now added in the text and documents cited in the reference list  (lines 290-292)

Who guideline was indeed already included.

Reviewer 2 Report

The paper is very interesting, however I see the following major issues that should be resolved before publishing this paper:

1.       Before objectives description should be explain the investigation question;

2.       On line 78 (method section) should have reference at the start date of study. Only, says “before 26/02/22”.

3.       The references 35,36,37 and 38 there are not on paper.

Author Response

The paper is very interesting, however I see the following major issues that should be resolved before publishing this paper:

thanks a lot for your positive comments

Before objectives description should be explain the investigation question;

done lines 145-146

  1. On line 78 (method section) should have reference at the start date of study. Only, says “before 26/02/22”

correcte frpm jan 2010 to june 2022

  1. The references 35,36,37 and 38 there are not on paper

we apologize amended

Reviewer 3 Report

The paper aimed to report the quality and the heterogeneity of a number of CPGs released from 2000 until May 30, 2022, which should provide clinicians with the most up-to-date and objective clinical evidence. The authors were using the AGREE II tool to test the quality of the CPGs despite the fact that since 2018 its update under the name AGREE-REX is also available. Please explain your decision to use AGREE II.

Statistical analysis was performed using Excel software - in the case of such an extensive comparative study, it would be more appropriate to use more sophisticated statistical software.

Despite the presented heterogeneity of recommendations, the greatest advantage of the presented work is the quality assessment of the CPGs themselves, which should help in the decision-making process and in the clinical practice, which guidelines should be further recommended.

The added value of the publication is summarizing the advantages and pointing out the shortcomings of CPGs, especially in terms of different or incomplete recommendations of vitamins, especially folic acid, as well as micronutrients.

Please consider deleting the word "gain" when using the abbreviation "GWG" which itself stands for gestational weight gain.

Author Response

The paper aimed to report the quality and the heterogeneity of a number of CPGs released from 2000 until May 30, 2022, which should provide clinicians with the most up-to-date and objective clinical evidence. The authors were using the AGREE II tool to test the quality of the CPGs despite the fact that since 2018 its update under the name AGREE-REX is also available. Please explain your decision to use AGREE II.

Good point thamk you. Now clarified on the basis of similar experience in nutrition (lines 140-142)

Statistical analysis was performed using Excel software - in the case of such an extensive comparative study, it would be more appropriate to use more sophisticated statistical software.

used  also  SPSS 

Despite the presented heterogeneity of recommendations, the greatest advantage of the presented work is the quality assessment of the CPGs themselves, which should help in the decision-making process and in the clinical practice, which guidelines should be further recommended.

at present unfortunately none and as stated in the conclusion there is a " need for CPGs shared by the different national and international body societies aimed at providing homogenous recommendations regarding nutrition and weight gain in pregnancy, also considering geographical differences and diet plans among the different populations"

The added value of the publication is summarizing the advantages and pointing out the shortcomings of CPGs, especially in terms of different or incomplete recommendations of vitamins, especially folic acid, as well as micronutrients.

thanks

Please consider deleting the word "gain" when using the abbreviation "GWG" which itself stands for gestational weight gain.

done thanks for the suggestion